# Quantitative Breast Density in Contrast-Enhanced Mammography

**DOI:** 10.3390/jcm10153309

**Published:** 2021-07-27

**Authors:** Gisella Gennaro, Melissa L. Hill, Elisabetta Bezzon, Francesca Caumo

**Affiliations:** 1Breast Radiology Department, Veneto Institute of Oncology (IRCCS), 35128 Padua, Italy; elisabetta.bezzon@iov.veneto.it (E.B.); francesca.caumo@iov.veneto.it (F.C.); 2Volpara Health Technologies Ltd., Wellington 6011, New Zealand; Melissa.Hill@volparasolutions.com

**Keywords:** breast density, contrast-enhanced mammography, mammography, tomosynthesis

## Abstract

Contrast-enhanced mammography (CEM) demonstrates a potential role in personalized screening models, in particular for women at increased risk and women with dense breasts. In this study, volumetric breast density (VBD) measured in CEM images was compared with VBD obtained from digital mammography (DM) or tomosynthesis (DBT) images. A total of 150 women who underwent CEM between March 2019 and December 2020, having at least a DM/DBT study performed before/after CEM, were included. Low-energy CEM (LE-CEM) and DM/DBT images were processed with automatic software to obtain the VBD. VBDs from the paired datasets were compared by Wilcoxon tests. A multivariate regression model was applied to analyze the relationship between VBD differences and multiple independent variables certainly or potentially affecting VBD. Median VBD was comparable for LE-CEM and DM/DBT (12.73% vs. 12.39%), not evidencing any statistically significant difference (*p* = 0.5855). VBD differences between LE-CEM and DM were associated with significant differences of glandular volume, breast thickness, compression force and pressure, contact area, and nipple-to-posterior-edge distance, i.e., variables reflecting differences in breast positioning (coefficient of determination 0.6023; multiple correlation coefficient 0.7761). Volumetric breast density was obtained from low-energy contrast-enhanced spectral mammography and was not significantly different from volumetric breast density measured from standard mammograms.

## 1. Introduction

The term “breast density” indicates the proportion of fibroglandular to fatty breast tissue that can be derived from mammography imaging. It has been proven that the clinical performance of mammography screening decreases as breast density increases, limiting the cancer detection rate while increasing the number of interval cancers [1,2]. Furthermore, breast density has gained increasing attention as a breast cancer risk factor [3,4], and is considered a potential individual biomarker to be included in breast cancer predicting models [5,6,7,8,9]. Due to this twofold role of breast density as masking and risk factor, there is a strong debate to change the “one fits all” mammography screening model into “personalized” screening models, i.e., screening programs including supplemental imaging for women with dense breasts or at increased risk for breast cancer [10,11,12].

Breast density can be evaluated using either human- or computer-based methods. Human-derived breast density is usually assessed by means of categorical variables, for instance, BI-RADS breast density [13], but is unavoidably affected by inter- and intra-observer variability [14,15]. There are several computer-based methods using semi-automatic or fully automatic algorithms, capable of computing breast density as a percentage of breast area or breast volume [16,17,18]. Volumetric breast density (VBD), i.e., the percentage volume of fibroglandular tissue in the whole breast volume as computed by automatic software tools measured from digital mammograms, has been included in statistical models used to estimate individuals’ risk of breast cancer [9,19]. Mammography is progressively being replaced by digital breast tomosynthesis (DBT); therefore, the same tools have been adapted to calculate VBD from tomosynthesis projection images [20].

Contrast-enhanced mammography (CEM, also called contrast-enhanced digital mammography, CEDM, or contrast-enhanced spectral mammography, CESM) is a dual-energy subtraction technique [21,22,23] applied to breast imaging [24,25,26], which has already produced initial results in screening contexts [27,28,29,30,31], showing its potential in personalized screening models [32,33]. In CEM, a pair of low-energy (LE) and high-energy (HE) images are acquired after contrast administration and used to construct the final “subtraction” image. The LE-CEM image is used to derive the morphological information, as for a standard mammogram, while the subtraction image provides functional information enhancements in areas of contrast uptake by possible breast lesions [24,25,26]. The prospective of using CEM as a detection tool requires that breast density can be obtained directly from CEM images. In this study, volumetric breast densities measured from LE-CEM images obtained within a study population of women at increased breast cancer risk were compared with those measured from mammography or tomosynthesis acquired before or after the CEM exam. The purpose of the study was to determine whether quantitative breast density derived from LE-CEM can be successfully used to feed breast cancer risk models within personalized screening protocols.

## 2. Materials and Methods

### 2.1. Study Design and Population

This observational cohort study (single-center) was approved by the institutional Ethics Committee. The study population included women enrolled from March 2019 to December 2020 in a prospective clinical trial (CE IOV #2017/92) comparing the clinical performance of CEM with breast MRI in a population of 300–500 women at intermediate and high risk for breast cancer. Signed informed consent, together with a questionnaire to gather the information required by the Tyrer–Cuzick breast cancer risk model, was obtained from all women enrolled in the prospective trial. For women without known BRCA1 or BRCA2 mutations, the lifetime risk was calculated using IBIS free software which implements v. 8.0 of the Tyrer–Cuzick model [34]; according to the NICE Guideline on familial breast cancer, enrolled women were classified at high risk for breast cancer if their lifetime risk was above 30%, and at intermediate risk if their lifetime risk was between 17% and 30% [35].

Subject selection for this observational study was exclusively based on the availability of mammography or DBT acquired before or after CEM. A total of 150 women with previous/subsequent mammography or DBT were used in this analysis.

### 2.2. CEM and Mammography/Tomosynthesis Imaging

Contrast-enhanced mammography was performed by a GE Senographe Pristina unit (GE Healthcare, Milwaukee Wis) after injection of a 1.5 mL/kg iodinated contrast agent (GE Omnipaque 350) using an automatic injector (3.0 mL/s). For pre-menopausal or peri-menopausal women, the examination was timed according to the phase of the woman’s menstrual cycle to minimize potential MRI false positives and possible background parenchymal enhancement. A two-view (cranio-caudal, CC, and medio-lateral oblique, MLO) bilateral examination was performed, starting two minutes after the contrast agent injection. For each mammography view, an image pair was acquired: (1) a low-energy image (LE-CEM) obtained using the same spectra as used for a standard mammography, such that most x-ray photons had energies below the absorption peak of the contrast agent (33.2 keV for iodinated contrast); (2) an HE image (HE-CEM) using copper filtration and tube voltage so that the resulting spectrum included photons mostly above the iodine k-edge. As previously mentioned, the LE-image and a contrast-enhanced subtraction image obtained from recombining LE-CEM and HE-CEM were used for diagnosis [24,25,26].

The subtracted CEM images were visually assessed by an experienced radiologist for contrast agent uptake in the normal tissue, or background parenchymal enhancement (BPE), and assigned one of four categories: minimal, mild, moderate, or marked [36].

The DM/DBT examinations were collected from among the conventional exams that were closest in time to the acquisition date of the CEM study. This may have been prior to or following the CEM exam date. Mammography examinations were acquired by one of the following digital units: GE Senographe DS, GE Senographe Essential, GE Senographe Pristina (the same equipment used for CEM), Hologic Selenia Dimensions, IMS Giotto Image 3DL, and Siemens Mammomat Inspiration. Tomosynthesis examinations were obtained either by GE Senographe Pristina or by Hologic Selenia Dimensions.

### 2.3. Breast Density

In this study, VBD was calculated by Volpara v. 1.5.5.1 (Volpara Health Ltd., Wellington, New Zealand) in a research mode to process CEM images. The Volpara algorithm uses a model of the physics of digital mammography to work backwards from the unprocessed image pixel value to the amounts of fibroglandular and adipose tissues that would result in the measured X-ray attenuation at each detector element location. The total breast volume is generated using the compressed breast thickness reported in the image DICOM header and a model for the shape of the breast. Using this estimated breast volume and the fibroglandular tissue volume derived from the X-ray attenuation, the VBD value (fibroglandular tissue volume/volume of breast) was calculated [17]. For CEM examinations, VBD computing was limited to the LE images. To help explain potential sources of density variability, other parameters studied for their potential association with VBD are breast thickness and compression force (both obtained from the image DICOM header), area of the compression paddle in contact with the breast [37], distance between the nipple and the posterior-edge (both determined by image analysis) [38], compression pressure calculated as the ratio between compression force and contact area, and mean glandular dose (MGD) obtained by applying the dosimetry model proposed by Dance and colleagues [39,40,41].

### 2.4. Statistical Analysis

VBD values obtained from LE-CEM and DM/DBT paired views were compared using a two-tailed Wilcoxon test. *p* < 0.05 was considered statistically significant. The same test was applied to any other variable listed above certainly or potentially affecting VBD values. Correlations between VBD measured from LE-CEM views and VBD measured from previous/subsequent DM/DBT views were evaluated by estimating the Spearman correlation coefficient, whereas the level of agreement between the two VBD datasets was explored thorough Bland–Altman plots. The same analysis (correlation and Bland–Altman) was performed for each patient case, after having averaged paired VBD values from available views (CC and MLO) for the two datasets.

Finally, after having obtained two paired per-patient datasets by averaging per-view VBDs and any other considered variable, the absolute difference was calculated between mean VBD obtained from CEM and previous/subsequent DM/DBT, as well as between any mean differences between all the other variables. A multiple regression model was then applied to analyze the relationship between the dependent variable “VBD difference” and the differences between any other variable considered independent. The regression model was weighted for 1/(VBD difference variance) to correct for heteroscedasticity.

Per-case VBD difference was also evaluated as a function of BPE assessed from CEM. VBD differences between the two subgroups showing minimal or mild BPE and moderate or marked BPE were compared with Mann–Whitney tests for independent samples, using the same 0.05 significance level.

Statistical analysis was performed using MedCalc^®^ Statistical Software version 19.7 (MedCalc Software Ltd., Ostend, Belgium).

## 3. Results

### 3.1. Study Population

Volumetric breast densities from 150 CEM examinations were compared with digital mammography or tomosynthesis performed either before or after CEM. On average, prior DM/DBT data were obtained 10.8 months before or after the CEM examination; the median time interval was 12 months, and 91% of cases (134/150) had a mammography or tomosynthesis ±15 months before/after CEM.

Table 1 shows the characteristics of the study population, including age, menopausal status, risk category, and breast density category.

The mean age was 51.0 ± 8.8 years, ranging between 35 and 76 years. Pre-menopausal women were 45.33% (68/150) of enrolled women, while the remaining 54.67% (82/150) were either peri- (14/150 = 9.33%) or post-menopausal (68/150 = 45.33%). Most of the subjects enrolled in the study were high-risk women (128/150 = 85.33%); 44.00% (66/150) had proven BRCA1 or BRCA2 mutations, and 41.33% (62/150) had a family history of breast cancer which, together with other risk factors also including VBD, led to a lifetime risk for breast cancer of above 30%. The remaining 14.67% of women (22/150) had intermediate risk (lifetime risk between 17% and 30%). Women included in the study population had mostly dense breasts: 76.00% (114/150) were classified as BI-RADS C or D, and 24.00% (36/150) were classified as BI-RADS A or B. Regarding the parenchymal background enhancement evaluated with CEM, 72.67% of women showed minimal or mild BPE (109/150), and 27.3% were moderate or marked (41/150). Previous or subsequent exams compared to CEM were 80% (120/150) DM and 20% (30/150) DBT.

### 3.2. Volumetric Breast Density

Table 2 shows results from the Wilcoxon paired test for VBD and all the other considered variables measured from LE-CEM and DM/DBT paired views.

Median VBD was comparable for LE-CEM and previous/subsequent DM/DBT (12.73% vs. 12.39%), not evidencing any statistically significant differences (*p* = 0.5855). Conversely, median differences between any other variable pairs were statistically significant (*p* < 0.05).

Figure 1 shows the correlation plot (on the left) and the Bland–Altman plot (on the right) for VBD measured from LE-CEM and previous/subsequent DM/DBT exams, for single views (upper plots) and for individual cases, obtaining average VBD values from multiple views (bottom plots).

The correlation was strong for both per-view and per-case comparisons (r = 0.87 and r = 0.92, respectively). In per-case analysis, the correlation was slightly improved compared to per-view analysis because some VBD differences for specific views were attenuated by averaging across multiple views belonging to the same study. The same effect was confirmed by the Bland–Altman plot: in both per-view and per-case plots, the mean VBD difference was zero, but the limits of agreement were narrower in per-case than in per-view analysis (about ±6% against ±8%). From the per-view plots, it can be observed that the largest VBD differences between LE-CEM and previous/subsequent DM/DBT images predominantly occurred for MLO views. The heat map in the per-case regression plots shows that volumetric breast density values were mostly below 15%. The VBD difference between LE-CEM and previous/subsequent DM/DBT tended to increase with breast density.

### 3.3. Multiple Regression Analysis

Least squares multiple regression (weighted for variance) using the per-case dataset showed that breast density variability between CEM and mammography/tomosynthesis was affected by all variables which can be considered to reflect differences in breast positioning, with the exclusion of breast volume. The sample case in Figure 2 shows the LE-CEM images of a woman with large fatty breasts in the upper part, and the mammograms acquired 12 months before in the lower part: there are large differences in breast volumes between the two examinations (CEM: 2634 cm^3^; DM: 2058 cm^3^; difference 576 cm^3^) associated with large differences in breast positioning (CEM mean nipple-to-posterior-edge distance: 174 mm; DM: 156 mm; 18 mm difference), not producing significant variation in volumetric breast density (CEM: 1.9%; DM: 2.7%). In contrast, the second sample case in Figure 3 shows LE-CEM images in the upper part, and the subsequent (11.5 months later) mammography of a woman with dense breasts in the lower part; mean VBD changes were from 20.1% to 30.5% because of better positioning in the second exam. In a subsequent mammography, breasts were better positioned, including about 1 cm more (nipple-to-posterior-edge distance) compared to CEM, resulting in a larger volume of fibroglandular tissue (136 cm^3^ vs. 64 cm^3^ with CEM) and larger overall breast volume (450 cm^3^ vs. 319 cm^3^ with CEM).

As reported in Table 3, a VBD difference between LE-CEM and DM/DBT was associated with significant differences of glandular volume, breast thickness, compression pressure, nipple-to-posterior-edge distance, and to differences of compression force and contact area, while the *p*-values for breast volume and MGD difference were above 0.05. The coefficient of determination (R^2^) was 0.6023, while the multiple correlation coefficient was 0.7761.

Figure 4 provides a scatter plot matrix of the VBD differences and the differences between all the independent variables measured in the paired LE-CESM and DM/DBT cases. A scatter plot matrix is a grid (or matrix) of scatter plots used to visualize bivariate relationships between combinations of variables. Each scatter plot in the matrix visualizes the relationship between a pair of variables, allowing many relationships to be explored in one chart. In the upper diagonal, the distribution (histogram) of each variable difference is represented.

The only parameter with which the VBD difference could be considered more than very weakly correlated is the difference in volume of glandular tissue (r = 0.545). Other correlations with moderate-to-strong relationships are recognizable between differences in breast volume and compressed breast thickness (r = 0.708), between breast volume differences and nipple-to-posterior-edge distance (r = 0.667), and finally between compression pressure differences and compression force differences (r = 0.880).

The Mann–Whitney test for independent samples applied to the subgroup of minimal or mild BPE compared with the subgroup of moderate or marked BPE did not show any statistically significant difference (*p* = 0.1197).

## 4. Discussion

The comparison between volumetric breast density measured in LE-CEM and in previous/subsequent DM or DBT images for the same patients did not find any statistically significant difference. This result was confirmed by the high correlation between the two VBD pairs for both per-view and per-case datasets, and with the Bland–Altman plots showing that the absolute differences between VBDs measured in LE-CEM and in DM/DBT were very close to zero, with limits of agreement moving from ±8% in per-view analysis to ±6% in per-case analysis, respectively. This result suggests that VBD measured from LE-CEM images is comparable with VBD obtained from standard mammography or tomosynthesis images. Therefore, if CEM is used as a detection tool, VBD measured using CEM can be used for risk evaluation as if it would be obtained from mammography. This reinforces the substantial clinical equivalence between LE-CEM and mammography images, as published by Lalji and colleagues, who compared LE-CEM and DM image quality by applying EUREF clinical criteria [42].

The multiple regression model showed that VBD differences are associated with variables related to breast compression (compression force and pressure, compressed breast thickness, and contact area) and with variables related to breast positioning (volume of glandular tissue and nipple-to-posterior-edge distance). X-ray breast imaging, including mammography, tomosynthesis, and CEM requires manual breast compression and positioning by a breast radiographer; therefore, it is very difficult, despite the application of criteria of “correct positioning”, to ensure that breast compression and positioning are the same laterality and view for consecutive exams. In other words, despite the known advantages of using quantitative tools to evaluate breast density (compared to subjective categorization affected by intra- and inter-observer variability for the same exam), radiologists should be aware that the reproducibility of breast density in two consecutive examinations can be subordinated to breast compression and positioning reproducibility. This topic was previously explored by Alonzo-Proulx et al. in a small study of repositioning the left breasts of 30 volunteers for a second CC view to evaluate the effect on measured density [43]. That study found a comparable VBD variability (between −4.25% and 2.28%) to that observed here, even in the context of repositioning the breast on the same day and imaging on the same equipment. Such awareness of the potential sensitivity of density measurement to breast positioning is particularly important in cases where the quantitative breast density is used as a decision-making index to drive supplemental imaging or risk assessments in personalized screening programs. Breast density has been shown to increase the accuracy of breast cancer risk models [9], although variability in breast density measures associated with changes in breast compression and positioning might have an impact on the predicted breast cancer risk. An investigation of breast positioning and compression quality is out of the scope of this work, but it is postulated that good breast positioning quality and consistent compression practices could mitigate the variability in automated density measurement, and subsequently, the impact on breast cancer risk estimates.

Apart from breast positioning changes, the only other physical means by which the quantitative density on LE-CEM could be different from DM/DBT should be related to the potential presence of contrast agents in the breast tissue, and their influence on the image pixel magnitudes. BPE assessed on the subtracted image was used here as a surrogate estimate for the amount of normal tissue contrast agent taken up. The findings demonstrated no association between BPE and VBD differences; therefore, it is estimated that the amount of iodine taken up in typical CEM produces signal intensity changes that are small enough as to not influence the density measurement method applied in this work.

The first implication of the study results is that in cases where CEM will be confirmed as a valid alternative to mammography and breast MRI in women at increased risk of breast cancer, CEM could be replaced by mammography with or without the addition of MRI (not used as a supplemental tool). For this reason, obtaining the same breast density values as standard mammography/tomosynthesis is useful and can be employed for risk assessments.

In addition to this direct implication, volumetric breast density may be a surrogate marker for response to neo-adjuvant chemotherapy (NAC), as reported by Engmann et al. (2017) [44]. Although responses to NAC can also be monitored with CEM [24,25,26,45], it will be of interest for future work to determine whether changes in VBD can provide independent and complementary information changes in contrast-enhancement on CEM. Furthermore, younger high-risk women who may benefit from CESM screening are also a target population for prevention strategies. Mammographic VBD and measurements of fibroglandular volume have been demonstrated as useful markers of breast density change associated with interventions that include chemoprevention [46,47,48], oophorectomy [49], diet [50], and weight loss [51].

This study has some limitations: the sample size was relatively small because for practical purposes, all available cases that met study criteria were included from the ancillary prospective trial comparing CEM and MRI performance among women at intermediate and high risk for breast cancer. As such, the study sample size was determined in an opportunistic manner, rather than being selected according to a power calculation. It is recommended that a larger sample be used in future work to confirm the results observed here. In addition, the inclusion of women not represented in the study population is encouraged, such as women at low risk for breast cancer. All LE-CEM images were produced by one type of equipment. If other vendor systems have different sensitivities to contrast agents, if a different injection protocol is applied, or if a substantially different compression is used, these results may not be applicable. Similar studies using other vendor systems are recommended. The study DM images were not obtained at the same time as the CEM images, and were often acquired using different imaging systems that are used for CEM. It is known that a woman’s breast density can change over time for a variety of reasons [52]. Nevertheless, the time interval between DM and CEM examinations (mean: 11 months; median: 12 months) was short enough to assume the substantial temporal stability of breast density, at least at a population level [52]. The use of different imaging systems between DM and CEM exams may actually have a greater influence on the variability of density results. For example, it is known that a change in compression paddle type can influence the amount of tissue in the field of view [53], and the combination of machine/paddle/compression modes can influence the accuracy of compressed thickness readout [54], both of which can influence the VBD estimate accuracy [55]. Only one automated density measurement tool was used in this study, which was a research-specific version compatible with LE-CEM. At the time of writing, we are not aware of any other automated density tools available for use with CEM images, in either a research setting or otherwise. In future work, it will be interesting to evaluate other density measurement software for this application, especially to test those with alternative measurement methods to understand the importance of the approach to any potential sensitivity to the presence of contrast agent in the breast tissue.

## 5. Conclusions

In conclusion, volumetric breast density can be obtained from contrast-enhanced spectral mammography (low-energy images), and is not significantly different from volumetric breast density values measured from standard mammograms, outside the inherent uncertainty associated with breast compression and positioning. This result will become particularly helpful if contrast-enhanced mammography gains a role as a key test in the personalization of screening programs for specific populations of women.

## Figures and Tables

**Figure 1 jcm-10-03309-f001:**
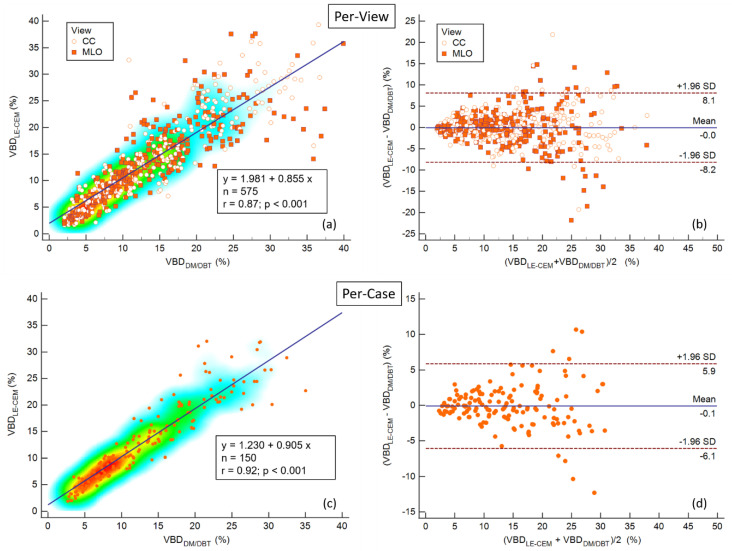
(**a**) Correlation plot of VBD measured in paired views obtained from LE-CESM and previous/subsequent DM/DBT exams; the Pearson’s correlation coefficient was r = 0.87, a reduction in correlation can be observed as VBD increases, especially for MLO views. (**b**) Bland–Altman plot of VBD differences between LE-CESM and previous/subsequent DM/DBT for each paired view; the mean difference was zero, the limits of agreement were ±8%. (**c**) Correlation plot of mean VBD obtained by averaging single-view VBDs for each paired study obtained with LE-CESM and DM/DBT; the Pearson’s correlation coefficient was r = 0.92, the heat map shows that VBD values were mostly grouped below 15%. (**d**) Bland–Altman plot of VBD differences between LE-CESM and previous/subsequent DM/DBT for each paired study; the mean difference was confirmed to be very close to zero, and the limits of agreement were reduced to ±5%.

**Figure 2 jcm-10-03309-f002:**
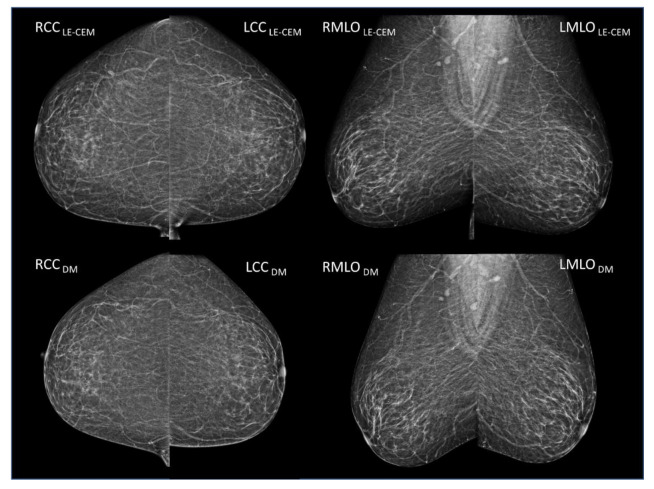
(Upper row) LE-CEM images of a woman with large fatty breasts. VBD: 1.9%; breast volume: 2634 cm^3^; volume of glandular tissue: 51 cm^3^; breast thickness: 85 mm; nipple-to-posterior-edge distance: 174 mm. (Lower row) DM images acquired 12 months before the CEM exam. VBD: 2.7%: breast volume: 2058 cm^3^; volume of glandular tissue: 54 cm^3^; breast thickness: 76 mm; nipple-to-posterior-edge distance: 155 mm. Despite the visible difference in breast positioning, the large reduction in breast volume due to worse positioning in the DM exam did not produce a significant variation in breast density, due to the breasts being predominantly fatty.

**Figure 3 jcm-10-03309-f003:**
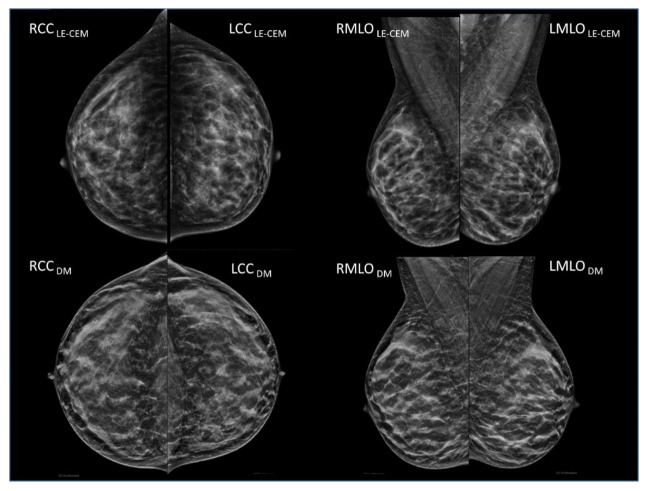
(Upper row) LE-CEM images of a woman with dense breasts. VBD: 20.1%; breast volume: 319 cm^3^; volume of glandular tissue: 64 cm^3^; breast thickness: 32 mm; nipple-to-posterior-edge distance: 86 mm. (Lower row) DM images acquired 11.5 months after the CEM exam. VBD: 30.5%; breast volume: 450 cm^3^; volume of glandular tissue: 136 cm^3^; breast thickness: 41 mm; nipple-to-posterior-edge distance: 98 mm. Breast positioning was better in the DM exam, including an additional volume of glandular tissue, leading to a significant increment in VBD.

**Figure 4 jcm-10-03309-f004:**
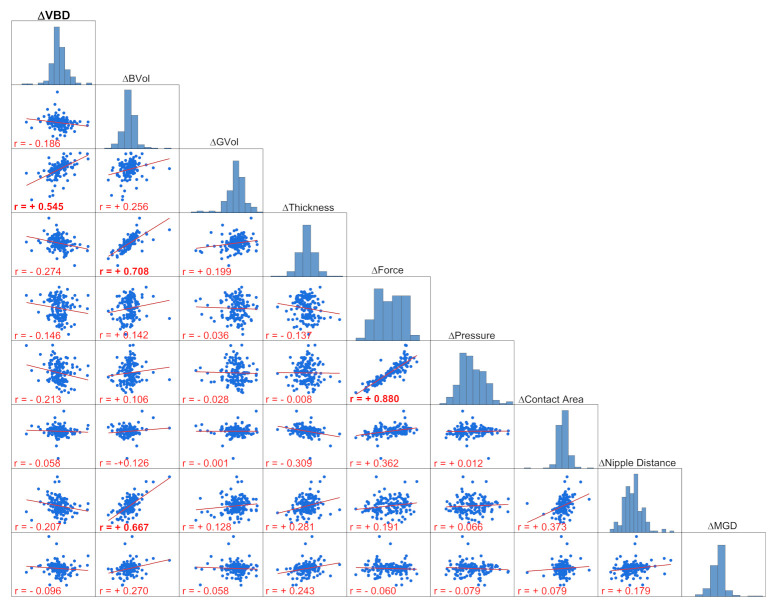
Scatter plot matrix of the VBD differences obtained from CEM and DM/DBT images and their relationship with all the other variable differences considered as independent in the multiple regression model: breast volume, volume of glandular tissue, breast thickness, compression force and pressure, contact area, nipple-to-posterior-edge distance, and mean glandular dose.

**Table 1 jcm-10-03309-t001:** Characteristics of study population.

Characteristic	Subgroup	Values
Number of women	/	150
Age (years)	mean ± SD median range	51.0 ± 8.8 51 (35–76)
Menopausal status	Premenopausal Perimenopausal Postmenopausal	68 (45.33%) 14 (9.33%) 68 (45.33%)
Risk category	BRCA1 ^1^ BRCA2 ^2^ HIGH ^3^ INTERMEDIATE ^4^	30 (20.00%) 36 (24.00%) 65 (41.33%) 22 (14.67%)
BI-RADS breast density ^5^	Predominantly fatty (A) Scattered fibroglandular (B) Heterogeneously dense (C) Extremely dense (D)	9 (6.00%) 27 (18.00%) 55 (36.67%) 59 (39.33%)
Background parenchymal enhancement (BPE)	Minimal Mild Moderate Marked	65 (43.33%) 44 (29.33%) 38 (25.33%) 3 (2.00%)
Previous/subsequent exam	Mammography Tomosynthesis	120 (80.00%) 30 (20.00%)

^1^ Women with a mutation of the BRCA1 gene; ^2^ women with a mutation of the BRCA2 gene; ^3^ lifetime risk ≥ 30%; ^4^ lifetime risk between 17% and 30%; ^5^ BI-RADS Breast Imaging-Reporting and Data System.

**Table 2 jcm-10-03309-t002:** Comparison between CEM and DM/DBT of volumetric breast density and related variables.

Variable	LE-CEM Median	DM/DBT Median	Hodges–Lehmann Median Difference	95% CI	*p*-Value
VBD (%)	12.73	12.39	0.075	−0.19 to 0.34	0.5855
Breast volume (cm^2^)	508.15	534.57	25.58	18.94 to 32.68	<0.0001
Glandular volume (cm^2^)	55.90	59.44	3.315	1.975 to 4.660	<0.0001
Breast thickness (mm)	47.7	50.0	2.35	1.90 to 2.80	<0.0001
Compression force (N)	102	85	−15.0	−17.5 to −12.0	<0.0001
Compression pressure (kPa)	11.83	10.46	1.655	−1.995 to 1.310	<0.0001
Contact area (mm^2^)	8529.46	8311.63	−144.14	−222.61 to 68.77	0.0003
Nipple distance from posterior edge (mm)	88.40	88.80	0.60	0.10 to 1.15	0.00180
Mean glandular dose (mGy)	1.544	1.527	−0.043	−0.074 to −0.012	0.0083

Median values obtained from LE-CEM and DM/DBT paired views, Hodges–Lehmann median difference and 95% confidence intervals, and *p*-values from the Wilcoxon paired test for VBD and all available variables actually or potentially associated with VBD. CEM, contrast-enhanced mammography; DM, digital mammography; DBT, digital breast tomosynthesis; VBD, volumetric breast density.

**Table 3 jcm-10-03309-t003:** Coefficients of the regression equation and *p*-values resulting from the multiple regression model.

Independent Variable Difference	Regression Coefficient	*p*-Value
(Constant)	0.1741	
Breast volume (cm^3^)	0.005017	0.1311
Glandular volume (cm^3^)	0.1174	<0.0001
Breast thickness (mm)	−0.2748	<0.0001
Compression force (N)	0.05008	0.0036
Compression pressure (kPa)	−0.5031	0.0001
Contact area (mm^2^)	−0.0009143	<0.0001
Nipple-to-posterior-edge distance (mm)	−0.1453	0.0011
Mean glandular dose (mGy)	−0.1373	0.6684

VBD differences between LE-CEM and DM/DBT were considered the dependent variable; differences in breast volume, glandular volume, breast thickness, contact area, compression force, compression pressure, nipple-to-posterior-edge distance, and mean glandular dose were included in the model as independent variables. CEM, contrast-enhanced mammography; DM, digital mammography; DBT, digital breast tomosynthesis; VBD, volumetric breast density.

## Data Availability

The data presented in this study are available on request from the corresponding author.

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
