# Peer review of "Quantitative Breast Density in Contrast-Enhanced Mammography"

_jcm, 2021, doi:10.3390/jcm10153309_

Round 1
Reviewer 1 Report
Aim : comparing the results of automated volumetric breast density measured in low energy contrast enhanced mammography (study group) vs digital mammography/tomography (control).
The study is an ancillary study of a prospective clinical trial comparing CEM and MRI.
Results : no statistical difference was shown
Methodology : OK (please mention how was determined the number of patient to include to prove the statistical equivalence between the two groups)
Manuscript is very clear, well-written.
Figures are comprehensive.
Potential causes of difference between CEM and mammography -based measures are cleverly adressed.
My only significant concern would be the real clinical implications of the study results. Indeed, the authors justify their research by highlighting the role of contrast enhanced mammography as an efficient supplemental screening modality proposed to patients with dense breasts. Hence, breast density should be determined at baseline, before any supplemental screening : once contrast enhanced mammography is decided, there is little interest for a second determination of breast density. That said, it can be of interest for research purpose or for longitudinally reassess the individual breast cancer risk.
Author Response
Response to Reviewer 1 Comments
Point 1: Methodology - Please mention how was determined the number of patients to include to prove the statistical equivalence between the two groups
Response 1:
In the Materials and Methods section 2.1, we explain that the subject selection was done opportunistically rather than according to a power calculation as follows, “Subject selection for this observational study was exclusively based on the availability of mammography or DBT acquired before or after CEM.”
However, we agree that it is important to also clarify that power analysis for subject selection was not done, which is a valid limitation of this work.
This comment has now been addressed within the Discussion section on the study limitations as follows:
“This study has limitations: the sample size was relatively small because for practical purposes, all available cases that met study criteria were included from the ancillary prospective trial comparing CEM and MRI performance among women at intermediate and high risk for breast cancer. As such, the study sample size was determined in an opportunistic manner, rather than being selected according to a power calculation. It is recommended that larger sample be used in future work to confirm the results observed here. In addition, inclusion of women not represented in the study population would be encouraged, such as women at low risk for breast cancer.”
Point 2: My only significant concern would be the real clinical implications of the study results. Indeed, the authors justify their research by highlighting the role of contrast enhanced mammography as an efficient supplemental screening modality proposed to patients with dense breasts. Hence, breast density should be determined at baseline, before any supplemental screening: once contrast enhanced mammography is decided, there is little interest for a second determination of breast density. That said, it can be of interest for research purpose or for longitudinally reassess the individual breast cancer risk.
Response 2:
We thank the reviewer for this comment and agree that the paper was missing some clinical context. In fact, Reviewer 2 made a similar comment. We felt it was most appropriate to address this concern by an expanded Discussion section with the following additions to put the study results in context:
“The first implication of study results is that in case CEM will be confirmed a valid alternative to mammography and breast MRI in women at increased risk of breast cancer, CEM would be replaced to mammography with or without the addition of MRI (not used as supplemental tool). For this reason, obtaining the same breast density values as standard mammography/tomosynthesis is useful and can be used for risk assessment.
In addition to this direct implication, volumetric breast density may be a surrogate marker for response to neo-adjuvant chemotherapy (NAC), as reported by Engmann et al. [44]. Although response to NAC can also be monitored with CEM [24-26, 45] it will be of interest for future work to determine whether changes in VBD can provide independent and complementary information changes in contrast-enhancement on CEM. Furthermore, younger high-risk women that may benefit from CESM screening are also a target population for prevention strategies. Mammographic VBD and measurements of fibroglandular volume have been demonstrated as useful markers of breast density change associated with interventions that include chemoprevention [46-48], oophorectomy [49], diet [50], and weight loss [51].”
Reviewer 2 Report
In this study, the authors investigated whether density measurement tools are also functioning on the low-energy images of a CEM examination. The study is compact and simple in design but addresses a very important topic: are density measurements also possible on CEM images. From prior studies, there are no reasons to think otherwise, but sometimes studies that state the obvious are desperately needed. Therefore, I think this topic has an important relevance to the breast imaging community.
Introduction:
Sound. Personally, a little too long regarding the focus of the study. It contains some mini reviews on the performance of CEM etc., which is rather irrelevant for the current topic. The main question is: do density measurements work? With this in mind, the authors might want to restructure the introduction a little bit.
Methods:
Figure 1 is not very useful. The techniques has been around since 2011. It is not required to explain the physics behind it. You could refer to other papers for more detail.
Results:
Sound, detailed analyses.
Discussion:
Study limitations are too limited. Yes, the size is too small, and this is somewhat of a concern, but this section should also address how this might influence results. Also, measurements are only done with Volpara, so we don’t know the results of other tools. Also, the DM and DBT were performed on different machines (quite a lot). In summary, the study limitations need to be extended: what are true limitations, what might their affect be on the results and could you have prevented them and if yes, why didn’t you do so? Does the population selection influence density results?
For the remainder, the discussion is sound but to my opinion it misses out on a certain amount of reflection with an eye on clinics or screening: what do these results mean for the imagin
Author Response
Response to Reviewer 2 Comments
Point 1: Introduction - Sound. Personally, a little too long regarding the focus of the study. It contains some mini reviews on the performance of CEM etc., which is rather irrelevant for the current topic. The main question is: do density measurements work? With this in mind, the authors might want to restructure the introduction a little bit. ps
Response 1: The Introduction was slightly reduced to improve the focus of this section.
Point 2: Method - Figure 1 is not very useful. The techniques has been around since 2011. It is not required to explain the physics behind it. You could refer to other papers for more detail.
Response 2: Figure 1 was removed. Indeed several of the cited papers already introduce the physics of the technique.
Point 3: Discussion - Study limitations are too limited. Yes, the size is too small, and this is somewhat of a concern, but this section should also address how this might influence results. Also, measurements are only done with Volpara, so we don’t know the results of other tools. Also, the DM and DBT were performed on different machines (quite a lot). In summary, the study limitations need to be extended: what are true limitations, what might their affect be on the results and could you have prevented them and if yes, why didn’t you do so? Does the population selection influence density results?
Response 3:
We agree with the reviewer that the limitations section could be expanded and appreciate the suggested topics. This Disucssion section has now been substantially expanded with further limitations as follows:
“This study has limitations: the sample size was relatively small because for practical purposes, all available cases that met study criteria were included from the ancillary prospective trial comparing CEM and MRI performance among women at intermediate and high risk for breast cancer. As such, the study sample size was determined in an opportunistic manner, rather than being selected according to a power calculation. It is recommended that larger sample be used in future work to confirm the results observed here. In addition, inclusion of women not represented in the study population would be encouraged, such as women at low risk for breast cancer. All LE-CEM images were produced by the one type of equipment. If other vendor systems have different sensitivity to contrast agent, or if a different injection protocol is applied, or a substantially different compression is used, these results may not be applicable. Similar studies using other vendor systems are recommended. The study DM images were not obtained at the same time as the CEM images, and were often acquired using different imaging systems that that used for CEM. It is known that a woman’s breast density can change over time for a variety of reasons [52]. Nevertheless, the time interval between DM and CEM examinations (mean: 11 months; median: 12 months) was short enough to assume substantial temporal stability of breast density, at least at a population level [52]. The use of different imaging systems between DM and CEM exams may actually have a greater influence on the variability of density results. For example, it is known that a change of compression paddle type can influence the amount of tissue in the field of view [53], and the combination of the machine/paddle/compression mode can influence the accuracy of compressed thickness readout [54], both of which can influence the VBD estimate accuracy [55]. Only one automated density measurement tool was used in this study, which was a research-specific version compatible with LE CEM. At the time of writing we are not aware of other automated density tools available for use with CEM images, in either a research setting or otherwise. In future work it will be interesting to evaluate other density measurement software for this application, especially to test those with alternative measurement methods to understand the importance of the approach to any potential sensitivity to the presence of contrast agent in the breast tissue.”
Point 4: Discussion - For the remainder, the discussion is sound but to my opinion it misses out on a certain amount of reflection with an eye on clinics or screening: what do these results mean for the imaging…
Response 4: The last sentence was truncated but the suggestion seems to be to better highlight the clinical impact of study results. If this was the case, we agree with this helpful comment. To address this concern, and also to address a similar suggestion from Reviewer 1, we have expanded the Discussion section with the following addtions to put the study results in better context for their clinical importance:
“The first implication of study results is that in case CEM will be confirmed a valid alternative to mammography and breast MRI in women at increased risk of breast cancer, CEM would be replaced to mammography with or without the addition of MRI (not used as supplemental tool). For this reason, obtaining the same breast density values as standard mammography/tomosynthesis is useful and can be used for risk assessment.
In addition to this direct implication, volumetric breast density may be a surrogate marker for response to neo-adjuvant chemotherapy (NAC), as reported by Engmann et al., 2017 [44]. Although response to NAC can also be monitored with CEM [24-26, 45] it will be of interest for future work to determine whether changes in VBD can provide independent and complementary information changes in contrast-enhancement on CEM. Furthermore, younger high-risk women that may benefit from CESM screening are also a target population for prevention strategies. Mammographic VBD and measurements of fibroglandular volume have been demonstrated as useful markers of breast density change associated with interventions that include chemoprevention [46-48], oophorectomy [49], diet [50], and weight loss [51].”